# Analysis of Engineering Characteristics and Microscopic Mechanism of Red Mud–Bauxite Tailings Mud Foam Light Soil

**DOI:** 10.3390/ma15051782

**Published:** 2022-02-26

**Authors:** Xiaoduo Ou, Shengjin Chen, Jie Jiang, Jinxi Qin, Zhijie Tan

**Affiliations:** 1College of Civil Engineering and Architecture, Guangxi University, Nanning 530004, China; ouxiaoduo@163.com; 2Guangxi Xinfazhan Communication Group Co., Ltd., Nanning 530029, China; qinjinxi1995@163.com; 3Guangxi Hualan Geotechnical Engineering Co., Ltd., Nanning 530004, China; wgytzj@163.com

**Keywords:** red mud, bauxite tailings mud, foamed lightweight soil, microscopic characteristics, environmental protection

## Abstract

In order to effectively utilize aluminum industrial waste—red mud and bauxite tailings mud—and reduce the adverse impact of waste on the environment and occupation of land resources, a red mud–bauxite tailings mud foam lightweight soil was developed based on the existing research results. Experiments were conducted to investigate the mechanical properties and microscopic characteristics of the developed materials with different proportions of red mud and bauxite tailings mud. Results show that with the increase in red mud content, the wet density and fluidity of the synthetic sample was increased. With 16% red mud content, the water stability coefficient of the synthetic sample reached its maximum of 0.826, as well as the unconfined compressive strength (UCS) of the sample cured for 28 d (1.056 MPa). SEM images reveal that some wastes of the sample without red mud were agglomerated, the peripheral hydration products were less wrapped, and when the amount of red mud was 16%, the hydration products tightly wrapped the waste particles and increased the structural compactness. The final concentration of alkali leaching of samples increased with the addition of red mud. The maximum concentration of alkali leaching was 384 mg/L for the group with the addition of red mud of 16%. Based on the obtained mechanical strength and alkali release analysis, the sample B24R16 was selected as the optimum among all tested groups. This study explored a way to reuse aluminum industrial waste, and the results are expected to be applied to roadbed and mining filling.

## 1. Introduction

Red mud and bauxite tailings mud are aluminum industrial wastes. Bauxite tailings mud is the water-bearing residue of bauxite raw ore (shallow clay ore) after grinding and washing [1]. Bauxite tailings mud is a kind of waste mud without alkali pollution. On the other hand, red mud is an alkaline solid waste produced during the production of alumina from bauxite [2,3]. As red mud contains soluble alkali [4,5], the leaching of soluble alkali under the action of precipitation will cause pollution to the surrounding environment, which seriously limits the application of red mud in engineering, resulting in an extremely low utilization rate of red mud. Therefore, in the process of utilization of red mud, in addition to considering mechanical properties, durability, and other conditions, the impact of the release of soluble alkali in red mud into the surrounding environment is of great significance. With the increase in global alumina production capacity, large amounts of aluminum industrial waste are produced every year [6,7], and aluminum production enterprises need a lot of land to store aluminum industrial waste, which has a huge impact on land resources and the natural environment [8,9]. How to reduce the occupation of land resources by waste and how to reduce the impact of waste on the surrounding environment become a top priority, and waste reuse would be an environmentally friendly way. Scholars have carried out multichannel resource utilization research on aluminum industrial waste.

The application of red mud in building materials is currently a research hotspot in red mud reuse and an important way to absorb red mud [10,11,12]. However, the high alkalinity of red mud is a major reason why it is difficult to use [7], so in the process of reusing red mud, attention should be paid to the solidification of soluble alkali in red mud and its impact on the environment. It was found that when red mud, fly ash, and desulfurized gypsum are used to prepare road base materials, in the case of (CaO + Na_2_O)/(SiO_2_ + Al_2_O_3_) = 0.88, the UCS of road base materials meets the strength requirements of highways. Additionally, the ion leaching concentration meets the drinking water standard [13]. Red mud mixed with 12% Ca(OH)_2_ and 1% gypsum has the best solidification effect, and the permeability of the mixture is much lower than 5.0 × 10^−8^ cm/s, such mixed material can be widely used in road base [14]. Soil enzymatic modification improves the strength and durability of red mud, which can be applied to geotechnical engineering fields, such as road embankments [15]. The findings mentioned above show that the modified red mud is of good engineering performance and can meet the requirements of engineering application and environmental protection.

However, few studies have been found in the literature regarding the application of bauxite tailings mud in engineering. It has been reported that bauxite tailings mud and blast furnace slag roasting can be used to prepare geopolymers [16], efficient CaO-based adsorbents can be prepared by using cheap CaCO_3_ and bauxite tailings mud [17], and bauxite tailings mud and eggshells can be used as raw materials to synthesize porous CaO-based adsorbents by solid-phase method [18]. In general, it is far away to consume a large amount of bauxite tailings mud by the current utilization methods. Since bauxite tailings mud contains a large amount of kaolin, it has good viscosity and plasticity [19,20]. Moreover, bauxite tailings mud is nontoxic and harmless, it is suggested using such waste as filling materials instead of fine sand and gravel in engineering fields, such as subgrade and mining area filling, thus increasing the utilization rate of the waste.

Currently, bauxite tailings mud and red mud wastes are commonly stored in aluminum mining enterprises, making it easy to access and low cost. If the feasibility of combining bauxite tailings mud and red mud to develop building materials is realized, it would make sure the reuse of the two wastes and further reduce environmental hazards.

The research of Peng et al. (2019) and Ou et al. (2020) revealed that when a small amount (0–20%) of bauxite tailings mud was added to make foamed lightweight soil, its strength and durability could be well satisfied. On the other hand, when a large amount (30–40%) of bauxite tailings mud was added, its strength and durability deteriorated seriously [1,21]. By considering that the soluble alkali in red mud can promote the hydration reaction [22], in this study, different contents of red mud were added in the bauxite tailings mud foam soil to replace the corresponding proportion of bauxite tailings mud, aiming to improve the engineering performance of the foam soil. The mechanism of strength increase was revealed by microscopic means, and the release characteristics of alkali in light soil materials were studied. Based on the experimental results, the optimal dosages of red mud and bauxite tailings mud were finally determined.

## 2. Materials and Methodology

### 2.1. Materials

#### 2.1.1. Red Mud and Bauxite Tailings Mud

The red mud used in the experiment was Bayer red mud. Both red mud and bauxite tailings mud were taken from an aluminum company in Baise, China. The bauxite tailings mud was air-dried and ground to 0.5 mm fine slag before the test. The chemical components of red mud and bauxite tailings mud were tested by XRD (the test method is shown in Section 2.3.3) and are shown in Table 1. It is shown that red mud contains more Na_2_O, which can provide hydroxide radicals for the hydration process.

#### 2.1.2. Foaming Agent

According to the Chinese standard GB/T 51238-2018 [23], a polymer composite foaming agent was used in this test.

#### 2.1.3. Cement

The cement used was C42.5 ordinary Portland cement, which was purchased from a cement factory in Nanning, China. It has a specific surface area of 341 m^2^/kg and a density of 3.1 g/cm^3^.

### 2.2. Test Design and Specimen Preparations

The soluble alkali in red mud mainly exists in the forms of NaOH, NaCO_3_, NaHCO_3_, and so on [22], which can enhance the pH value in the slurry environment and promote the hydration reaction of the system. If red mud is incorporated into building materials, the reaction process in the system will be affected if the dosage is more than 20%, and the intuitive effect is the reduction of strength [24,25]. Combined with the research results of Peng et al. (2019) and Ou et al. (2020) [1,21], and according to the Chinese standard CJJ/T 177-2012 [24], under the condition of 700 kg/m^3^ wet density, it is designed to mix different amounts of red mud (the proportion of red mud is within 20%) in the foam light soil group mixed with 40% bauxite tailings mud to replace part of the bauxite tailings mud. The influence of red mud content on the physical and mechanical properties and environmental effects of bauxite tailings mud foam lightweight soil was analyzed. According to the Chinese standard CJJ/T 177-2012 [26], the amount of each material is calculated, and the proportion of each component of the foam light soil mixed with red mud bauxite tailings mud is shown in Table 2.

Note that in the serial ID, B is the abbreviation of bauxite tailings mud and R is red mud. Their following numbers represent the mass ratio (in dry weight) of the corresponding component in the cementitious material. The water–binder ratio refers to the mass ratio of water to cementitious material. To do so, a referenced mixture, labelled as B40R0 in Table 2, was prepared without adding red mud. In this mixture, the dry mass of aluminum bauxite tailings mud accounted for 40% in the cementitious material. Five other serials were manufactured with the mass ratio of red mud in the cementitious material of 4%, 8%, 12%, 16%, and 20% (Table 2).

Certain masses of red mud, bauxite tailings mud, cement, and water were weighted according to details in Table 2 and mixed evenly to obtain the mud body. Then, the foam made from a foaming agent was mixed with the mud body evenly to form aluminum industrial waste foam light soil. After the fluidity and wet density of the foamed lightweight soil were tested, the foamed soil was poured into a cubic mold (Figure 1). The foamed lightweight soil was covered with a plastic film and left to statically stand for 24 h under a controlled temperature of 20 °C. Afterwards, the mold was removed. The solidified foamed lightweight soil specimens were sealed and cured in an environmental simulation generator (relative humidity of 95 ± 3%, temperature of 22 ± 2 °C) before the test.

### 2.3. Experimental Scheme

#### 2.3.1. Fluidity and Wet Density Test

Mobility was tested according to the Chinese standard GB/T 2419-2005 [27]. Wet density was tested according to the Chinese standard JTG3430-2020 [28]. The combination in Table 2 was tested for fluidity and wet density.

#### 2.3.2. Unconfined Compression Test

For each serial listed in Table 2, unconfined compression tests were conducted on the well-prepared solidified specimens cured for 7 and 28 days, respectively. A WAW-600 microcomputer-controlled electrohydraulic servo universal testing machine (Fangyuan Testing Equipment Co., Ltd., Jinan, China) was used to measure the unconfined compressive strength of the specimens at various ages. The loading rate was 0.5 kN/s. Six samples were taken for each ratio, and six samples were taken at the end. The average value of the test results of the samples was taken as the compressive strength of the experimental material. In addition, in order to investigate the wetting–drying effect, extra specimens were prepared to experience five wetting–drying cycles after 28 days’ curing. The cured specimens were submerged in distilled water for wetting and then oven-dried under 105 °C for drying. The test program is detailed in Table 3.

#### 2.3.3. XRD Test

The mineral composition of foamed lightweight soil was tested by XRD, and the sample was dried and ground into powder before testing (Table 4). The mineral composition was qualitatively analyzed by an X’Pert PRO MRD/XL high-resolution diffractometer (PANalytical, Almelo, Netherlands). During the test, the ray wavelength λκα was 1.54060 Å, the tube pressure was 40 KV, the tube flow was 40 mA, the scanning range was 5°~75°, the step size was 0.02°, and the scanning speed was 5°/min. 

#### 2.3.4. SEM Test

The microstructure of foamed lightweight soil was tested by SEM; before the test, the samples were cut into 6 mm × 4 mm × 2 mm slices by a geotechnical knife and dried; and the samples were sprayed with gold for further use (Table 4). Microscopic feature analysis was carried out with an S-3400N scanning electron microscope (Hitachi Corporation, Tokyo, Japan), the magnification was between 20 and 300,000 times, the temperature of the sample observation chamber was set to 50 °C, and the pressure was controlled to 650 Pa.

#### 2.3.5. Determination of Na^+^

According to the Chinese standard GB50986-2014 [29], the alkali content in red mud was calculated by Na_2_O, so the alkali concentration was characterized by Na^+^ concentration. The leaching solution was extracted according to the Chinese standard GB7053-86 [30], and the Na + concentration was tested by flame photometer method according to the Chinese standard JJG 630-2007 [31].

## 3. Results and Discussion

### 3.1. Fluidity and Wet Density

Figure 2 shows the change of wet density and fluidity of each group with the red mud content.

It can be seen from Figure 2 that when the red mud content increased from 0% to 20%, the wet density showed an increasing trend, and the wet density was more than 700 kg/m^3^, higher than the designed wet density. The wet density of the group mixed with 20% red mud is the largest, reaching 760 kg/m^3^. The reasons for this phenomenon are as follows: the small particles of bauxite tailings mud and red mud are conducive to filling the gaps between foams and forming a relatively dense light soil system, thus increasing the wet density. The proportion of red mud is larger than that of bauxite tailings mud, so the wet density increases with the increase in red mud content.

The fluidity of foamed light clay in red mud also increases with the increase in red mud substitution. In the group without red mud (B40R0), the fluidity was 155 mm, indicating that the requirement that the fluidity should be controlled within 160 mm~200 mm cannot be met without admixture in accordance with the Chinese standard CJJ/T 177-2012 [26]. However, the fluidity of bauxite tailings mud was obviously improved after the red mud was replaced by bauxite tailings mud. When the amount of red mud was 4~20%, the fluidity was in the range of 168 mm~177 mm, which can meet the requirements of technical regulations without the addition of admixtures.

### 3.2. Unconfined Compressive Strength Test

Figure 3 shows the unconfined compressive strength test results of red mud foam lightweight soil samples prepared by mixing red mud with different contents after curing for 7 d and 28 d.

In the range of 0~16% red mud content, the unconfined compressive strength at 7 d and 28 d greatly increased with the increase in red mud content, and reached the maximum value of 0.872 MPa and 1.056 MPa, respectively, when the red mud content was 16%. With the continuous increase in red mud content, the strength decreased slightly. In the group B40R0 without red mud, its unconfined compressive strength at 7 d and 28 d was 0.298 MPa and 0.410 MPa, respectively. In the group B24R16 with 16% red mud content, its 7 d and 28 d compressive strength was 2.93 and 2.57 times that of the group B40R0. The strength of the group mixed with 16% red mud increased significantly.

Table 5 shows the influence of red mud with different incorporation amounts on early strength.

According to Table 5, the 7 d curing strength value of the B40R0, B36R4, B32R8, B28R12, B24R16, and B20R20 groups was 0.727, 0.713, 0.764, 0.744, 0.826, and 0.829 of the 28 d curing strength value. With the increase in red mud content, the early strength effect was obvious. The reason is that adding a certain amount of red mud containing alkali elements is beneficial to accelerate the process of hydration reaction in the system and achieve the required strength in a short time in engineering, which is conducive to the development of later construction.

Table 6 shows the unconfined compressive strength test results and the calculation results of water stability coefficient of foam lightweight soil with different red mud contents after five times of the dry–wet circulation after curing for 28 d.

Table 6 shows that the dry–wet circulation obviously reduced the unconfined compressive strength of foam lightweight soil, especially that the unconfined compressive strength of the B40R0 group after five dry–wet cycles was only 0.293 MPa, and the UCS loss rate was 0.269, which was the highest in all groups. The analysis reasons are as follows: bauxite tailings mud cannot completely hydrate with cement in foam light soil, and part of bauxite tailings mud only act as filler aggregate between foams. When the system is subjected to repeated dry–wet circulation, the bauxite tailings mud particles that have not undergone hydration reaction will absorb water and increase in volume, thus reducing the adhesion between them and the hydration products, which will easily fall off from the aggregate and affect the integrity of the products between foams, and finally make the strength deteriorate seriously after the dry–wet circulation. The UCS loss rate decreased to 0.181 when only 4% red mud was added into the foamed lightweight soil, indicating that the incorporation of red mud promoted the hydration reaction of the system and improved the durability of foamed lightweight soil. When the red mud content is 0–16%, the UCS loss rate of foamed lightweight soil decreases with the increase in red mud content. When the red mud content is 16%, the UCS loss rate is the smallest, which is 0.108, corresponding to the maximum UCS value.

### 3.3. XRD Analysis

When 4% red mud was added, the strength and durability of the sample were greatly improved, while when 16% red mud was added, the strength and durability can reach the maximum value. Therefore, according to the mechanical properties of the samples mixed with different red mud, three representative groups, B40R0, B36R4, and B24R16, were selected for XRD analysis.

Figure 4 shows the XRD patterns of hydrated 28 d solidified bodies for each group. The figures show that in each category of hydration, 28 d solidified the body in the spectrogram besides containing red mud, bauxite tailings mud quartz (SiO_2_), gibbsite (Al(OH)_3_), kaolinite (Al(Si_2_O_5_)(OH)_4_), hematite (Fe_2_O_3_) such as original phase, calcium hydroxide (CH, Ca(OH)_2_), calcium carbonate (CaCO_3_), ettringite (Aft, Ca_6_Al_2_(SO_4_)_3_(OH)_12_·26H_2_O), calcium silicate hydrate gel (C-S-H, CaO_x_·SiO_2_·H_2_O_y_), gismondine (CaAl_2_Si_2_O_8_·4H_2_O), and a series of hydration products. By comparing the XRD patterns of each group, it was found that the diffraction peak intensity of C-S-H and ettringite increased with the increase in red mud dosage, which was consistent with the phenomenon where UCS increased with the increase in red mud dosage in the UCS test. It is worth noting that the diffraction peak of Al(OH)_3_ in the B24R16 group disappears. It is speculated that the OH^−^ produced by cement hydration and the OH released by adding red mud together dissolve Al(OH)_3_, and the AlO_2_ produced after dissolution continues to form [Al(OH)_6_]^3−^ with the OH^−^ and water in the system. [Al(OH)_6_]^3−^ was further combined with Ca^2+^ and SO_4_^2−^ to form C-S-H and ettringite [32]. The resulting reaction equation is shown below.
Al(OH)_3_) + OH^−^ = AlO_2_^−^ + 2H_2_OAlO_2_^−^ + 2OH^−^ + 2H_2_O = [Al(OH)_6_]^3−^2[Al(OH)_6_]^3−^ + 3Ca^2+^ + (X − 6)H_2_O = Ca_3_Al_2_O_6_·XH_2_O2[Al(OH)_6_]^3−^ + 6Ca^2+^ + 3SO_4_^2−^ + 26H_2_O = Ca_6_Al_2_(SO_4_)_3_(OH)_12_·26H_2_O

### 3.4. SEM Analysis

Corresponding to the XRD test, three groups, B40R0, B36R4, and B24R16, were selected for curing for 28 days, and the microstructure and product shape were observed by scanning electron microscopy under conditions of 50 and 5000 times.

Figure 5 shows the 50-fold scanning electron microscope images of samples in the B40R0, B36R4, and B24R16 groups and the binarization images processed by Image-Pro Plus 6.0 software. As shown in the figure, the foam had a great influence on the pore structure inside the sample. A large number of foams caused a large number of pores inside the sample, and some pores were connected, which reduced the density of the sample and significantly weakened its strength. With the increase in the amount of red mud, the pore size of the foam increases, but the thickness of the cell wall between the pores increases with the increase in the amount of red mud, and the increase in the thickness of the cell wall is conducive to the development of strength. It is speculated that the soluble alkali in red mud promotes the hydration reaction, and the more hydration products are present, the more red mud and bauxite tailings mud particles are encapsulated, resulting in thicker cell walls and higher strength. Table 7 shows that the porosities of the samples in the three groups were 75.91, 77.82, and 78.73, respectively, which were very close, indicating that the replacement of partial bauxite tailings mud by red mud had little influence on the porosity of the sample structure, but it could increase cell wall thickness.

Figure 6, Figure 7 and Figure 8 are 5000-fold scanning electron microscope images of samples in the groups B40R0, B36R4, and B24R16. Fibrous calcium silicate hydrate (C-S-H), sheet or hexagonal slab Ca(OH)_2_, and a small amount of needle-like ettringite can be seen in each group. When cement, bauxite tailings mud, red mud, and other materials were mixed in each group, cement hydration reaction with water generated C-S-H, CH, Aft, and other cementation products. Meanwhile, Al_2_O_3_ and SiO_2_ rich in red mud and bauxite tailings mud reacted with Ca^+^ in volcanic ash to generate C-S-H, which improved the strength of the samples. The bauxite tailings mud content of the sample B40R0 group was the largest, and some waste particles agglomerated in the hydration process, and the agglomerates were only partially wrapped by hydration products outside the body, which made the strength of the sample low after curing for 28 d, which is also the reason why the unconfined compressive strength of the sample in this group was only 0.41 MPa. The sample B36R4 group added 4% of red mud because bauxite tailings mud mixed with still more. There were still some samples after particles in water clouds appeared, but because red mud contains many Na_2_O, can provide a hydration process with hydroxyl, and promotes the hydration process, to generate a more mottled gelling material wrap in most of the areas, the integral structure was made more compact. Therefore, the unconfined compressive strength of this group of specimens was nearly 50% higher than that of B40R0 specimens. No agglomeration particles were found in the B24R16 group samples, and a large number of C-S-H and CH generated by hydration reaction tightly wrapped the red mud and bauxite tailings mud particles in the system that were not involved in the hydration reaction. Moreover, as red mud was added to this group the most, part of the soluble basic ions in red mud accelerated the hydration reaction process, significantly increasing the density of the structure. Therefore, the UCS strength of the combination was the highest among all combinations, and the water stability coefficient was also the largest.

### 3.5. Analysis of Alkali Release Characteristics

The change in Na^+^ leaching concentration and time is shown in Figure 9. The concentration of leached alkali in each group was roughly the same, showing a law of first increasing, then decreasing and finally becoming stable. On the one hand, the addition of red mud promoted the hydration reaction of the system, and the hydration substance generated in the system had a stable curing effect on alkali [33,34]. However, with the addition of red mud, the amount of soluble alkali introduced increased, and the leaching amount increased. According to the Chinese standard GB/T 14848-2017 [35], the maximum alkali leaching concentration was 384 mg/L for groups with a red mud content of 16%. The limit value of the four types of water in the non-supergroundwater was 400 mg/L, and the amount of alkali leaching in the B20R20 group exceeded the limit value of the four types of water. In this case, the selection of this mix ratio will have an obvious impact on the environment. Combined with UCS analysis results, the B24R16 group had the highest UCS strength, and the alkali leaching amount of this group did not exceed the four water limits, so it belongs to the best group.

## 4. Conclusions

In this study, aluminum industrial waste was used to prepare foam lightweight soil from red mud–bauxite tailings mud. The test mainly analyzed the physical and mechanical properties, microscopic characteristics, and alkali environmental effects of foam lightweight soil from red mud–bauxite tailings mud, and explored a way to reuse aluminum industrial waste, the results of which can be applied to the needs of roadbed and filling. The main conclusions of this chapter are as follows:

When the red mud content increased from 0% to 20%, both wet density and fluidity increased with the increase in red mud content. When the red mud content was between 4% and 20%, the fluidity of the foam light soil of red mud–bauxite tailings mud was between 168 mm and 177 mm, which meets the requirements of the specification and is conducive to filling.

When the red mud content was 0–16%, with the increase in red mud content, the unconfined compressive strength of 7 d and 28 d samples increased greatly and reached the maximum values of 0.872 MPa and 1.056 MPa, respectively, when the red mud content was 16%, which increased 192.6% and 157.6% compared with the group without red mud content, respectively. However, the water stability coefficient increased with the increase in red mud content and reached the maximum value of 0.892 when the red mud content was 16%.

The microscopic test revealed that some wastes of the samples without red mud were agglomerated, and the surrounding hydration products were less wrapped, resulting in a lower structural strength of the samples. When the red mud content reached 16%, there were more hydration products, and the red mud and bauxite tailings mud particles that were not involved in the hydration reaction were tightly wrapped so that the sample structure was dense and the strength was the highest.

The final concentration of alkali leaching increased with the addition of red mud. When the addition amount was 16%, the concentration of alkali leaching was still lower than the four types of water limit specified, with a mixing amount of 20%. The maximum alkali leaching concentration was 460 mg/L, which exceeded the limit value of the four types of groundwater. According to UCS and alkali release analysis results, the B24R16 group had the highest UCS intensity, and the alkali leaching amount of this group did not exceed the four water limits, so it belongs to the optimal group.

## Figures and Tables

**Figure 1 materials-15-01782-f001:**
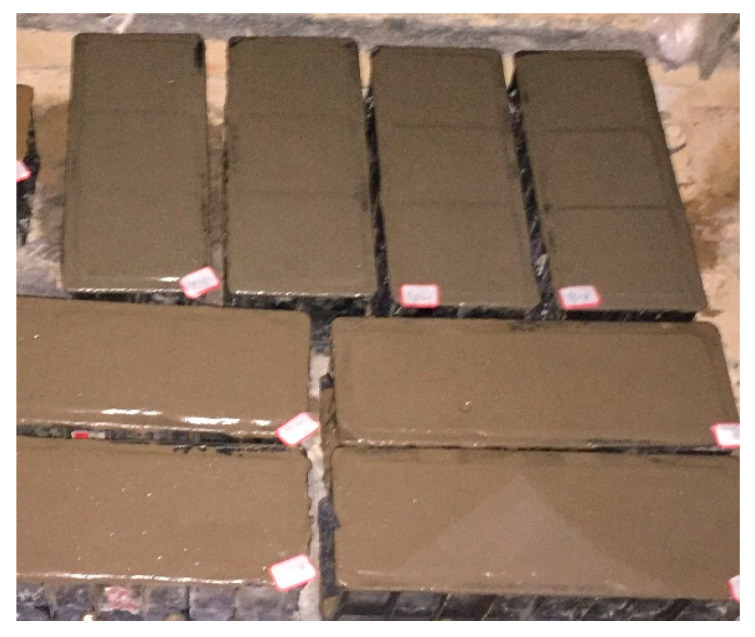
Injection mold for foam lightweight soil.

**Figure 2 materials-15-01782-f002:**
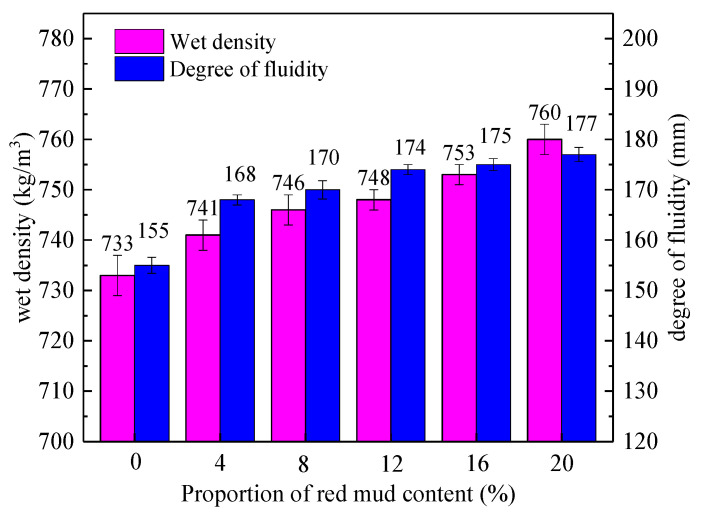
Variation of wet density and fluidity with red mud content.

**Figure 3 materials-15-01782-f003:**
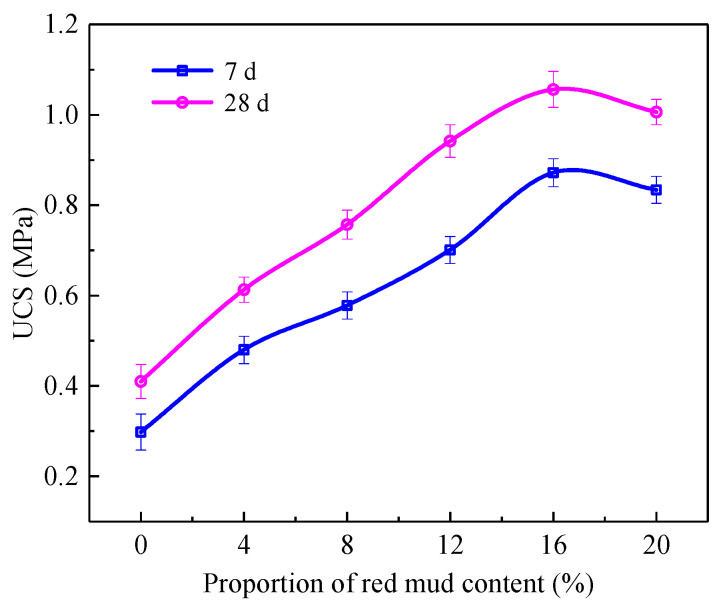
Variation of unconfined compressive strength with red mud content.

**Figure 4 materials-15-01782-f004:**
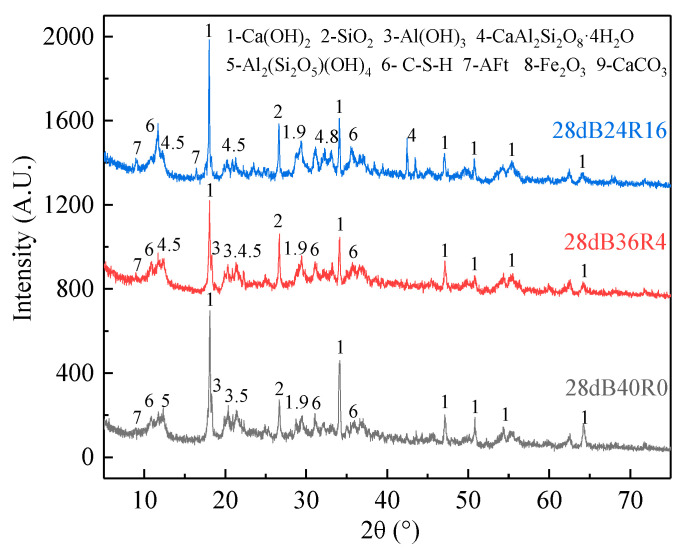
XRD patterns of B40R0, B36R4, and B24R16 cured for 28 days.

**Figure 5 materials-15-01782-f005:**

SEM image at 50× and binary image of B40R0, B36R4, and B24R16 specimens.

**Figure 6 materials-15-01782-f006:**
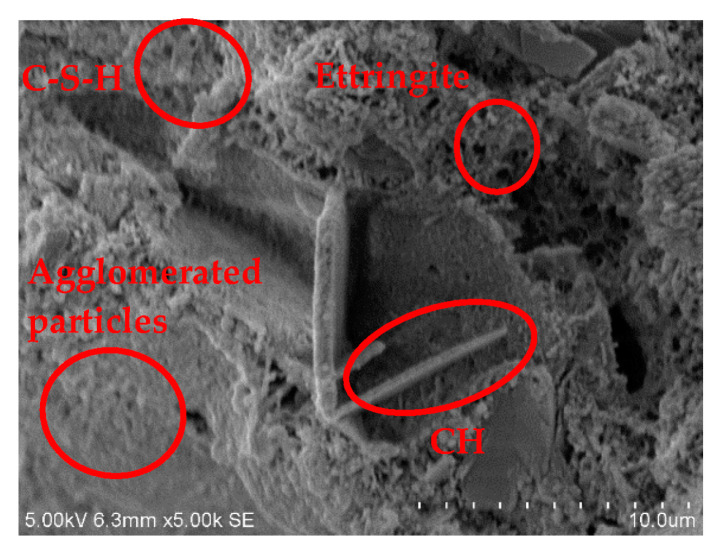
SEM image at 5000× of B40R0 specimen.

**Figure 7 materials-15-01782-f007:**
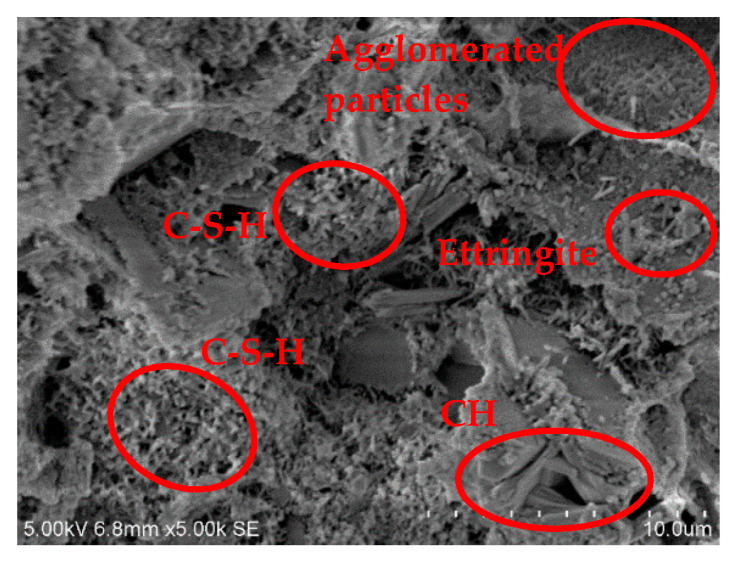
SEM image at 5000× of B36R4 specimen.

**Figure 8 materials-15-01782-f008:**
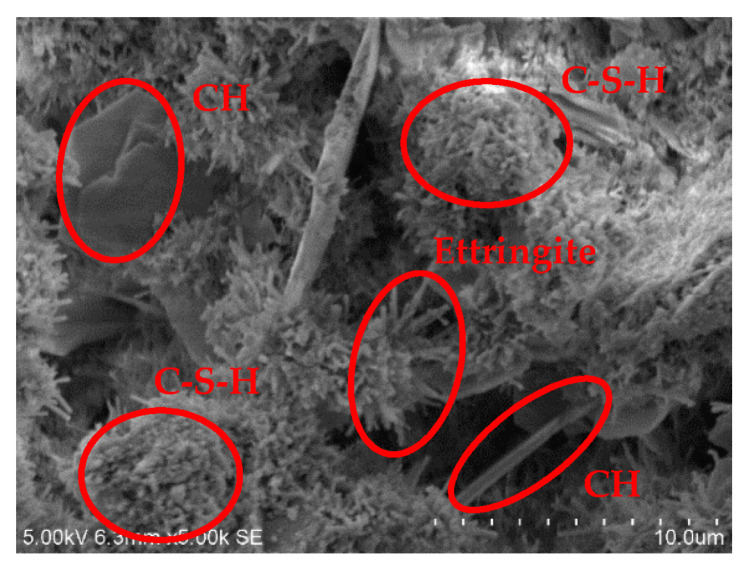
SEM image at 5000× of B24R16 specimen.

**Figure 9 materials-15-01782-f009:**
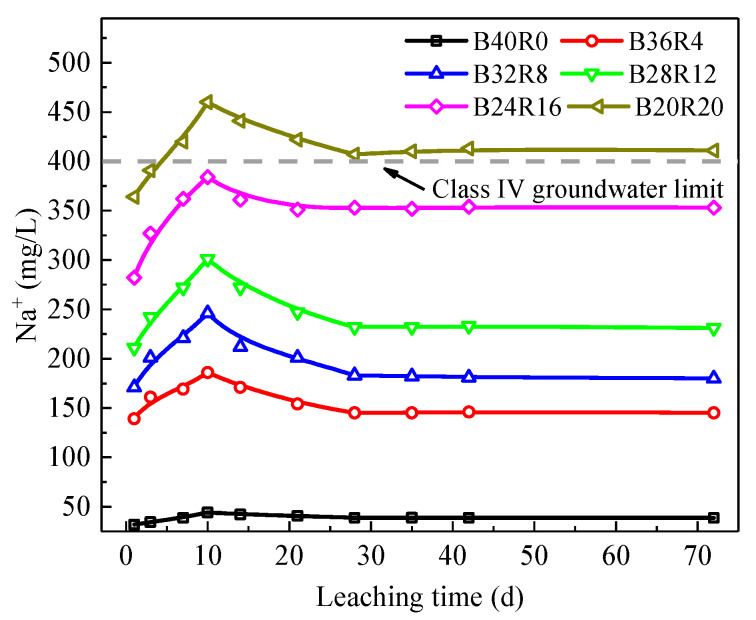
Variation diagram of alkali concentration in semidynamic toxicity leaching test solution.

**Table 1 materials-15-01782-t001:** Main mineral composition of mud (wt%).

Chemical Composition	Fe_2_O_3_	Al_2_O_3_	SiO_2_	CaO	Na_2_O	K_2_O	MgO	TiO_2_	MnO	LOI
Red mud	29.5	21.6	15.1	11.5	9.21	0.16	0.60	5.59	0.21	5.10
Bauxite tailings mud	14.93	38.10	28.32	0.32	/	0.78	/	1.70	/	3.02

**Table 2 materials-15-01782-t002:** Proportions of red mud and bauxite tailings mud of lightweight soil.

Serial ID	Total Cementitious Material (kg/m^3^)	Cement (kg/m^3^)	Proportion of Red Mud Content (% by Weight)	Red Mud (kg/m^3^)	Bauxite Tailings Mud (kg/m^3^)	Foam (L/m^3^)	Water–Binder Ratio
B40R0	420	252	0	0.0	168.0	602	0.6
B36R4	420	252	4	16.8	151.2	602	0.6
B32R8	420	252	8	33.6	134.4	602	0.6
B28R12	420	252	12	50.4	117.6	602	0.6
B24R16	420	252	16	67.2	100.8	602	0.6
B20R20	420	252	20	84.0	84.0	602	0.6

**Table 3 materials-15-01782-t003:** Specifications of the testing program for UCS.

Types of Test	Tested Mixture	Initial Water Content of Specimen (%)	Dimensions of Specimen (mm)	Curing Duration (Days)	Curing Conditions	Other Condition
Unconfined compression test	All mixture listed in Table 2	Optimum water content	Φ 50 × H 50	7, 28	20 °C, 95% humidity	/
Dry–wet circulation	All mixture listed in Table 2	Optimum water content	Φ 50 × H 50	28	20 °C, 95% humidity	Five wetting–drying cycles, oven-dried under 105 °C

**Table 4 materials-15-01782-t004:** Specifications of the testing program for XRD and SEM.

Types of Test	Tested Mixture	Dimensions of Specimen (mm)	Curing Duration (Days)	Process
XRD	B40R0, B36R4, and B24R16	Powdery	28	Dried the sample, ground the sample into powder
SEM	B40R0, B36R4, and B24R16	6 × 4 × 2	28	Sliced the sample, dried the sample, sprayed gold on the sample

**Table 5 materials-15-01782-t005:** Ratio of 7 d UCS to 28 d UCS.

Serial ID	B40R0	B36R4	B32R8	B28R12	B24R16	B20R20
Ratio	0.727	0.783	0.764	0.744	0.826	0.829

**Table 6 materials-15-01782-t006:** Dry–wet cycling test results of red mud foam lightweight soil.

Serial ID	Red Mud Content (%)	28 d UCS P_1_ (MPa)	After Five Times of the Dry–Wet Circulation of UCS P_2_ (MPa)	UCS Loss Rate (P_1_ − P_2_)/P_1_
B40R0	0	0.401	0.293	0.269
B36R4	4	0.613	0.502	0.181
B32R8	8	0.757	0.635	0.161
B28R12	12	0.942	0.823	0.126
B24R16	16	1.056	0.942	0.108
B20R20	20	1.006	0.887	0.113

**Table 7 materials-15-01782-t007:** Porosity of B40R0, B36R4, and B24R16 specimens.

Serial ID	B40R0	B36R4	B24R16
Porosity	75.91	77.82	78.73

## Data Availability

Some or all data, models, and code that support the findings of this study are available from the corresponding author upon reasonable request.

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
