# Peer review of "Analysis of Engineering Characteristics and Microscopic Mechanism of Red Mud–Bauxite Tailings Mud Foam Light Soil"

_materials, 2022, doi:10.3390/ma15051782_

Round 1

Reviewer 1 Report

The manuscript entitled " Analysis of Engineering Characteristics and Microscopic  Mechanism of Red Mud-Tailing Mud Foam Light Soil " reports a study on the creation of cementitious products using two industrial wastes from aluminum processing. The industrial wastes used are red mud and tailing mud. The authors' idea is that combination of the simultaneous use of these two industrial wastes. Six systems were created with different amounts of these wastes. The samples were characterized.

The subject matter is consistent with the purpose of the journal, although it does not show particular elements of originality. I believe that it can nevertheless be useful for researchers working in this area of research.

I believe the manuscript may be considered for publication after minor revision.

Here are some tips for authors.

Table 1 shows the percentages by weight of the minerals present in the sludge. How were these compositions researched? What does the remaining 7% of red mud bouxite contain? What does the remaining 15% of the Tailing mud contain? These percentages do not indicate, could they be heavy metals ?. This is very important to specify.

(Line 102) Can you explain why the foaming agent was used?

No specific indication of the instrumentation and experimental conditions is given. It is advisable to enter the description of the instrumentation used (eg. XRD, SEM.ecc)

Author Response

Response to Reviewer 1 Comments

  • Point 1: Table 1 shows the percentages by weight of the minerals present in the sludge. How were these compositions researched? What does the remaining 7% of red mud bouxite contain? What does the remaining 15% of the Tailing mud contain? These percentages do not indicate, could they be heavy metals ? This is very important to specify.

Response 1: Table 1 is supplemented and improved in the paper. The main chemical components of red mud and tailings mud are tested by x-ray diffraction. After the test, interpretation is carried out, the main products listed in the table have high peaks and are easy to interpret, a small amount of products have small peaks and are not easy to interpret. Test results and literature review show that there is no heavy metal pollution in red mud and tailing mud.

  • Point 2: (Line 102) Can you explain why the foaming agent was used?

Response 2: Foaming agent is a key material to increase material pores and reduce specific gravity, and is an important component of lightweight soil.

  • Point 3: No specific indication of the instrumentation and experimental conditions is given. It is advisable to enter the description of the instrumentation used (eg. XRD, SEM.ecc).

Response 3: Agree with the review comments, the manuscript will be improved according to the review comments.

Reviewer 2 Report

The authors report an experimental study on Characteristics and Microscopic Mechanism of Red Mud-Tailing Mud Foam Light Soil. Overall, the manuscript is well written.

Some comments for improving the paper are as follows:

  1. It is discussed that red mud promoted the hydration reaction of the system and improved the durability of foamed lightweight soil. However, the reason behind this is not discussed. Please discuss this with justification.
  2. XRD results can be supported by previous studies.
  3. Conclusion should not have any discussion.

Author Response

Response to Reviewer 2 Comments

  • Point 1: It is discussed that red mud promoted the hydration reaction of the system and improved the durability of foamed lightweight soil. However, the reason behind this is not discussed. Please discuss this with justification.

Response 1: The paper mentions that the addition of red mud promotes the hydration reaction of the system, which increases the C-S-H and ettringite in the light soil, thereby improving the strength and durability of the light soil.

  • Point 2: XRD results can be supported by previous studies.

Response 2: Yes, the interpretation of the XRD pattern refers to the previous results.

  • Point 3: Conclusion should not have any discussion.

Response 3: Agree with the review comments, the conclusions have been revised.

Reviewer 3 Report

  1. How the radio actitive material property in  redmud is identified? discuss
  2. What is the influence of Fe2O3 presence in the redmud?
  3. On what basis Table 2 was constructed?
  4. Photograph of the fabrication facility with its specification need to be incorporated 
  5. Any scentific reason to select 28 days curing? 
  6. What standard followed in compression test?
  7. Fig 5 Need more explanation

Author Response

Response to Reviewer 3 Comments

  • Point 1: How the radio actitive material property in red mud is identified? discuss.

Response 1: According to literature analysis, radioactive substances can be measured by gamma spectroscopy and elemental transformation.

  • Point 2: What is the influence of Fe2O3 presence in the red mud?

Response 2: Fe2O3 has no effect on the hydration process, but it can make the light soil material reddish.

  • Point 3: On what basis Table 2 was constructed?

Response 3: The manuscript will be improved according to the review comments.

  • Point 4: Photograph of the fabrication facility with its specification need to be incorporated?

Response 4: Agree with the review comments, the manuscript will be improved according to the review comments.

  • Point 5: Any scentific reason to select 28 days curing?

Response 5: With reference to relevant specifications and literature, cement-based cementitious materials refer to the 28-day curing standard.

  • Point 6: What standard followed in compression test?

Response 6: The Chinese standard of Technical specification for foamed mixture lightweight soil filling engineering (CJJ/T 177-2012).

  • Point 7: Fig 5 Need more explanation.

Response 7: Agree with the review comments, the manuscript will be improved according to the review comments.

Reviewer 4 Report

The article deals with an interesting and important topic of industrial waste management.

The authors narrowly formulated the topic, conducted laboratory research. They chose the methodology properly. The obtained results are correct.

Nevertheless, the article could be improved prior to publication.

  1. In the introduction, one should refer to more international literature items in order to provide a broader overview of the research conducted so far.
  2. The experience and results are valuable and interesting, but I suggest that their use be presented in a broader industrial and economic context (in summary). 

Author Response

Response to Reviewer 4 Comments

  • Point 1: In the introduction, one should refer to more international literature items in order to provide a broader overview of the research conducted so far.

Response 1: Agree with the review comments, the manuscript will be improved according to the review comments.

  • Point 2: The experience and results are valuable and interesting, but I suggest that their use be presented in a broader industrial and economic context (in summary).

Response 2: Agree with the review comments, the manuscript will be improved according to the review comments.

Round 2

Reviewer 3 Report

The paper can be accepted in the present form 

Author Response

Agree with the review comments, the manuscript will be improved according to the review comments.